# Genotoxicity and Oxidative Stress in Experimental Hybrid Catfish Exposed to Heavy Metals in a Municipal Landfill Reservoir

**DOI:** 10.3390/ijerph17061980

**Published:** 2020-03-17

**Authors:** Lamyai Neeratanaphan, Chuchart Kamollerd, Pimchanok Suwannathada, Pongthorn Suwannathada, Bundit Tengjaroenkul

**Affiliations:** 1Research Group on Toxic Substances in Livestock and Aquatic Animals, Khon Kaen University, Khon Kaen 40002, Thailand; hlamya@kku.ac.th (L.N.); Chukam@kku.ac.th (C.K.); nonmou@kku.ac.th (P.S.); ponxuw@kku.ac.th (P.S.); 2Faculty of Science, Khon Kaen University, Khon Kaen 40002, Thailand; 3Faculty of Veterinary Medicine, Khon Kaen University, Khon Kaen 40002, Thailand

**Keywords:** catfish, genotoxicity, landfill, metal, stress

## Abstract

This study aimed to investigate the concentrations of Cr, Cd and Pb in the water, sediment and experimental hybrid catfish muscles, and to compare the genetic differentiation and the levels of oxidative stress biomarkers (malondialdehyde and protein carbonyl) between the catfish from the contaminated reservoir near a municipal landfill and the reference area after chronic exposure. The concentrations of all metals in the water and the concentration of Cd in the sediment exceeded Thailand’s surface water quality and soil quality standards, respectively, whereas the concentrations of these metals in fish muscles did not exceed Thailand’s food quality standards. Dendrogram results in terms of genetic similarity values of the catfish from the reference and the landfill areas were 0.90 to 0.96 and 0.79 to 0.86, respectively, implying that the genetic differentiation of the fish from the landfill was greater than of those from the reference area. The fish in the landfill reservoir had slightly increased protein carbonyl levels. The results indicate that chronic heavy metal exposure can cause genotoxicity of the hybrid catfish and induce protein carbonyl as an oxidative stress biomarker in the reservoir near a municipal landfill.

## 1. Introduction

Municipal and commercial communities have rapidly and continuously expanded in Khon Kaen Province, Thailand [1]. This expansion has been accompanied by the increase of both municipal and commercial solid waste. Due to improper landfill waste management, toxic leachate that can spread heavy metals and other toxic substances around the Khon Kaen municipal landfill environment is present [2]. The major metal contaminants in the Khon Kaen municipal landfill are chromium (Cr), cadmium (Cd) and lead (Pb) [3,4,5,6,7]. Sriuttha et al. [7] demonstrated that concentrations of Cr, Cd and Pb in the water and sediment near the municipal landfill were 0.016 ± 0.009, not detected and 0.009 ± 0.0006 mg/L, and 19.91 ± 0.96, 0.47 ± 0.23 and 5.40 ± 0.13 mg/kg, respectively, whereas Intamat et al. [6] determined that arsenic (As) concentrations in the water, sediment and tilapia fish muscles in the landfill environment were 0.006 ± 0.002, 1.08 ± 0.64 and 0.16 ± 0.16 mg/kg, respectively. From previous reports, these toxic metals can spread from hazardous waste, such as metal containers, light bulbs and batteries, into leachate and surrounding reservoirs [8,9], and can be absorbed and accumulated in aquatic organisms, including fish [10]. Intamat et al. [6] and Sriuttha et al. [7] found that the Cr concentration in *Anabas testudineus* and *Rasbora tornieri* as well as the Cd concentrations in *Barbonymus gonionotus* and *Oreochromis niloticus* exceeded the limit of the international standard; additionally, Pb and As concentrations in all four Khon Kaen landfill fish species exceeded the international standards. Generally, fish living in toxic metal-contaminated reservoirs can absorb the metals through their gills, skin and digestive tract [2,11,12], which can lead to biomagnification, and can cause health risks to consumers [13]. 

Several research studies have demonstrated molecular biomarkers after heavy metal exposure in fish, particularly through the application of extensive molecular (including whole genome) approaches to test genotoxicity in fish, including the damage on chromosome and deoxyribonucleic acid (DNA) structures [8,9]. Furthermore, many reports have shown that heavy metals induce an imbalance between the production and reduction of free radical species, i.e., causing oxidative stress in fish [14,15]. The free radicals can attack both phospholipid cell membranes and intracellular protein molecules to induce the oxidative stress products, including increased protein carbonyl levels [16,17,18,19]. Recently, the types and amount of oxidative stress products have been considered as biomarkers for environmental indices in terms of both status and situation of aquatic ecosystems [20,21,22,23]. The effects of toxic metals on aquatic animals have been reported, but information on their consequences to DNA patterns and oxidative stress to fish after both acute and chronic exposure in landfill ecosystems is limited. Therefore, the aims of this study were to investigate heavy metal accumulation as well as to compare genetic differentiation and oxidative stress biomarkers after chronic exposure between the experimental hybrid catfish from the reservoirs near the municipal landfill and the reference area. The effects of heavy metals from municipal landfill leachate on the fish in this study can be used to help make laws, policies, standards and guidelines in collaboration with government and local administration agencies with sustainable ecosystem management at present and in the future.

## 2. Materials and Methods

### 2.1. Study Site

A reservoir near the Khon Kaen municipal landfill as the study site was located at latitude 16°35′41.30″ N and longitude 102°48′12.11″ E (Figure 1), and its geographic coordinate is UTM 48Q N 1,835,718 E 266,085 at a height of 200 m above sea level.

### 2.2. Fish and Experimental Design

Juvenile hybrid catfish (*Clarias gariepinus* x *C. macrocephalus*) weighing approximately 14 g were cultured at a density of 30 fish/m^2^ in 3 × 3 m in floating net cages in a reservoir that received leachate from the Khon Kaen municipal landfill. The experimental catfish were randomly divided into 2 groups (reference and landfill) with 3 replications of each group, and they were fed on commercial feed pellet (Charoen Pokphan Co., Thailand) at 9% by weight, 3 times per day for 4 months simulating the length of catfish production as in commercial fishery practice. At the end of the experiment, each catfish muscle was measured for heavy metal concentrations, the gills and liver were excised for genetic differentiation evaluation and blood plasma was collected from the tail vessels for the plasma oxidative stress test. The reference catfish were collected from the Inland Fisheries Research and Development Center, Khon Kaen, Thailand, which was set as the non-affected heavy metal contamination area. 

### 2.3. Water Quality Parameters

Water-dissolved oxygen, pH, temperature, total dissolved solids and electro-conductivity were measured at the experimental reservoirs using mobile digital meters at 9:00 a.m. Total ammonia nitrogen was measured using the standard titration method [24] (Table 1).

### 2.4. Heavy Metal Measurements

The collected water and sediment samples after chronic exposure for 4 months were digested following the Environmental Protection Agency (EPA) method 6010, and the Cr, Cd and Pb concentrations were analyzed using inductively coupled plasma optical emission spectrometry (ICP-OES) [25]. The detection limits of each analyzed metal were 0.001 mg/kg for Cr as well as Cd, and 0.005 mg/kg for Pb. The ICP-OES wavelengths for the Cr, Cd and Pb analyses were 226.502, 188.979 and 267.716 nm, respectively.

Analyses of blanks and standards were conducted at every 10th sample. The concentrations of the heavy metals in the procedural blanks were significantly <5% of the mean analyzed concentrations for all metals. Replications of the analyses were conducted to guarantee the precision and accuracy of all metal measurements. The results were found not to deviate by more than 2% of the certified levels [24]. The heavy metal recovery values were calculated by acceptance in the range of 85–115% [26]. The results were 90–100% of the acceptable values and considered as accurate.

### 2.5. Genotoxicity Study

The mixtures of extracted DNA molecules from the fish gills and livers were successfully amplified by 25 inter-simple sequence repeat (ISSR) primers (Table 2) in a PCR cycler (Flex Cycler^2^, Analytikjena) [27]. Each DNA band was evaluated and recorded as the following diallelic characters: present = 1 and absent = 0. All of the evaluated band results were transferred to the set dendrogram for genetic similarity/differentiation evaluation of the catfish from the studied areas [28].

### 2.6. Oxidative Stress Biomarkers

After being cultured in the landfill reservoir, the blood plasma of ten experimental catfish was collected to measure malondialdehyde (MDA) from thiobarbituric acid [29,30], and to measure protein carbonyl (PC) from its reaction with dinitrophenylhydrazine [30].

### 2.7. Statistical Analyses

For statistical analyses of the genotoxicity study, each evaluated DNA band was transferred to the set dendrogram by the NTSYSpc 2.1 program [28], and for the oxidative stress test, levels of the plasmas’ MDA and PC from the landfill and reference catfish were analyzed using the Mann–Whitney U-test. All of the statistical tests were conducted at a 95% confidence level.

## 3. Results

### 3.1. Water Quality Parameters

The water quality parameters of the reference and the landfill reservoir water samples after four months of the experiment are shown in Table 3.

### 3.2. Heavy Metal Concentrations

The heavy metal concentrations in the water, sediment and catfish muscles from the reference and landfill reservoirs are shown in Table 4 and Table 5.

### 3.3. Genotoxicity

The extracted DNA from the gills and livers of the catfish were mixed for genotoxicity evaluation. The 25 ISSR primers were succeeded to produce good quality DNA bands. Examples of ISSR fingerprints from the samples in the reference and landfill catfish are shown in Figure 2. The ISSR patterns generated 2613 total bands (ranging in size from 200 bp to 20,000 bp) with 352 characteristics, including 164 similar bands and 188 different bands. All bands were applied to the set dendrogram according to the Dice coefficient. The dendrogram results separated the DNA samples into two groups based on the studied areas. The first group was comprised of five reference catfish samples (1.1–1.5), and the second group was comprised of five landfill catfish samples (2.1–2.5) (Figure 3). The genetic similarity values among the individuals of the catfish from the reference and landfill areas are shown in Table 6. 

The genetic similarity values of fish samples ranged from 0.90 to 0.96 for samples 1.1–1.5, and from 0.79 to 0.86 for samples 2.1–2.5, respectively. Comparing data of genetic similarity values between each individual fish, the lowest genetic similarity value was 0.65 between the fish samples 1.4 and 2.4. On the other hand, the highest genetic similarity value was 0.96 between the fish samples 1.1 and 1.2 (Table 6).

### 3.4. Oxidative Stress Biomarkers

#### 3.4.1. Malondialdehyde

Levels of plasma MDA in the catfish are shown in Figure 4. The catfish from the reservoir near the Khon Kaen municipal landfill revealed a decrease in plasma MDA but no significant difference as compared with the reference catfish (*P* > 0.05).

#### 3.4.2. Protein Carbonyl

The levels of plasma PC in the catfish are shown in Figure 5. In comparison with the plasma PC of the reference fish (1.11 nmol/mg protein), the PC of the landfill fish demonstrated a slightly greater level (1.15 nmol/mg protein), but not significantly different (*P* > 0.05).

## 4. Discussion

### 4.1. Water Quality Parameters

The results of the water quality parameters of the reference and the landfill reservoir water samples after four months of the experiment (Table 3) were not significantly different, and were within the standards for fishery practices [24]. Water quality data revealed conditions of landfill leachate and its suitability for aquatic ecosystems. Low water quality can negatively induce stress, metabolism, reproduction, genetics and health of fish [34,35,36]. In this study, the water quality parameters from the reference and the landfill reservoirs were within Thailand’s standard limits for surface water sources [1,37]; therefore, genotoxicity and plasma oxidative stress consequences in chronically exposed catfish may not be as affected by the water quality parameters in the landfill reservoir. 

### 4.2. Heavy Metal Concentrations

The concentrations of Cr, Cd and Pb in the water and Cd concentration in the sediment exceeded Thailand’s water quality standards for surface water sources and Thailand’s soil quality standard, respectively [1]. The metal concentrations in the catfish muscles did not exceed Thailand’s food quality standards [33]. Management of waste in the Khon Kaen municipal landfill was not appropriate, resulting in the increase of municipal and hazardous metals, which can spread to leachate and can be potential sources of heavy metals to affect fish in the landfill reservoir [5]. Similar to other polluted landfills, heavy metal contamination in the Khon Kaen landfill leachate and sediment of the reservoir can be transferred and accumulated in aquatic organisms [10]. The bioaccumulation phenomena were identified by the reports of Intamat et al. [6] and Promsid [38], who demonstrated heavy metal accumulation in plants and fish, for example, *Monopterus albus*, *Channa striata* and *C. batrachus*. Intamat et al. [6] showed that As in landfill fish *B. gonionotus, R. tornieri*, *A. testudineus* and *O. niloticus* exceeded Thailand’s food quality standards. Sriuttha et al. [7] revealed that the Cr concentration in *A. testudineus* and *R. tornieri* as well as the Cd concentration in *B. gonionotus* and *O. niloticus* exceeded the limit of international standards (FAO, USA), and the Pb concentrations in four fish species exceeded the limit of international standards. Furthermore, this study investigated the Cr, Cd and Pb concentrations in the hybrid catfish as an important fish species consumed throughout local communities near the Khon Kaen municipal landfill. Generally, the hybrid catfish in the floating net cages in the landfill reservoir, after four months, could simulate the experimental condition as in fishery practices in Thailand. Because the experimental hybrid catfish had a relatively long period of exposure (four months), they could continuously accumulate heavy metals from leachate and sediments into their bodies [39,40,41,42]. The metal accumulations in the catfish probably occurred because they are a scaleless skin species with a large body surface area. Similar to several studies, chronic exposure of Pb and Cr in African catfish (*C. gariepinus*) may lead to stress, morphology changes in the gills and liver, ionic regulatory alterations, DNA damage as well as genetic polymorphism [42,43,44,45,46,47]. 

The fish that accumulate toxic heavy metals can do harm to consumers in local communities, inducing detrimental disorders and diseases related to these metals [48,49,50]. In general, the potential health risk from accumulated metals in the fish in the landfill reservoir to consumer health depends on several factors, including species, concentration, time and kinetics of the heavy metals contained in the exposed animals [51,52].

### 4.3. Genotoxicity

The ISSR patterns from 25 successful primers in this study produced 2613 total DNA bands with 188 different bands. The results of the dendrogram separated the fish DNA samples into two groups based on the studied areas. As the concentrations of Cr, Cd and Pb in the fish from the contaminated reservoir were higher than the fish from the reference area, the genetic differentiation values of the fish from the contaminated reservoir were greater than of the fish from the reference area. These findings suggest that the metal accumulations in the landfill fish could affect their genotoxicity in terms of the molecular nucleic acid profile. In contaminated environments, heavy metal exposure could demonstrate genotoxicity effects as DNA damages in fish as single- and double-strand breakages, alterations in DNA repair processes, oxidation of nucleic bases and DNA-protein crosslinks [19,53,54]. When heavy metals enter the cell membranes, they generally can cause genotoxicity through several mechanisms. For example, Cr exposure revealed modification of the DNA nitrogenous base, DNA strand breakage and protein-Cr-DNA adducts [18,55,56]. Cd can induce oxidative stress, ploidy changes, substitutions and oxidization in gene bases, DNA damage, mutagenesis, deletions and point mutations [16,57]. Pb can induce oxidative damages, alteration in gene transcription, mitogenesis, carcinogenic events involved in DNA damage and other indirect genotoxic changes [58]. Furthermore, the loss of DNA’s structural or functional integrity in exposed organisms can initiate deleterious effects at both the individual and population levels, especially through impaired growth or reproduction [58,59,60]. Therefore, experimental catfish could be a potential genotoxicity as bioindicator in aquatic ecosystems. In addition, local communities around the municipal landfill should increase their concern and enhance environmental management to reduce and prevent the risks to human health from consuming the heavy metals accumulated in the fish from the landfill reservoir.

### 4.4. Oxidative Stress Biomarker

#### 4.4.1. Malondialdehyde

Levels of plasma MDA in the catfish from the reservoir near the Khon Kaen municipal landfill revealed a decrease in plasma MDA and no significant difference as compared with the reference fish. Generally, contaminated landfill reservoir leachate and sediments are usually comprised of complex mixtures of heavy metals that accumulate in aquatic organisms [10]. Heavy metals can induce reactive oxygen and nitrogen species, i.e., causing free radical overload or oxidative stress in aquatic organisms, including fish [15]. Evaluation of oxidative damage in fish can directly reflect metal exposure in an aquatic environment, which could affect the health of the creatures [14]. Contaminated heavy metals in living fish generally interact with cell membrane phospholipid molecules, nuclear proteins and nuclear nucleic acids, resulting in increases in MDA and fragmentation of DNA strands [14,15].

MDA is a product derived from lipid peroxidation on cell membrane phospholipids and circulating lipids, and an increased level of MDA content directly involves a degree of oxidative injury from the toxic agents [61,62,63]. The flounder, catfish, killifish, mullet and stickleback revealed an increase in MDA after exposure to pollutants [64,65,66]. In this study, a slight decrease in plasma MDA levels was found in the hybrid catfish, indicating that the fish may possess strengthened cellular structures and/or metabolic processes as well as enzymes to protect themselves from the presence of heavy metals and oxidative stress products. Therefore, the hybrid catfish could be less affected from oxidative stress on cell pathology and homeostasis such as protein metabolism, glutathione pathways and mitochondrial functions than other fish species [67,68]. Presently, oxidative stressed fish have revealed different responses according to the types and concentrations of the pollutants as well as the habitat and feeding behavior of the fish [68].

#### 4.4.2. Protein Carbonyl

In comparison with the plasma PC of the reference fish (1.11 nmol/mg protein), the PC of the landfill fish demonstrated a slightly greater level (1.15 nmol/mg protein), but not significantly different. Generally, the occurrence of PC correlates with protein damages caused by oxidative stress, which has been demonstrated from a number of diseases and/or tissue lesions [69]; therefore, PC can be a marker as a result of the degradation of enzymes, the alterations of amino acid structures and the changes of protein functions [70,71,72,73]. In accordance with previous reports, the elevated level of PC in the catfish from the landfill reservoir could suggest that metal pollutants may cause protein damage as a consequence of oxidative stress. 

## 5. Conclusions

The concentrations of Cr, Cd and Pb in the experimental hybrid catfish after chronic exposure to heavy metals in a reservoir near a municipal landfill for four months did not exceed Thailand’s food quality standards. Compared to the fish in the reference area, the fish in the landfill area demonstrated a relatively greater genetic differentiation with no differences between both fish groups according to MDA and PC values. Results of the study can conclude that Cr, Cd and Pb can potentially induce genetic alterations, and there is no oxidative stress in the hybrid catfish from the reservoir near the Khon Kaen municipal landfill. 

## Figures and Tables

**Figure 1 ijerph-17-01980-f001:**
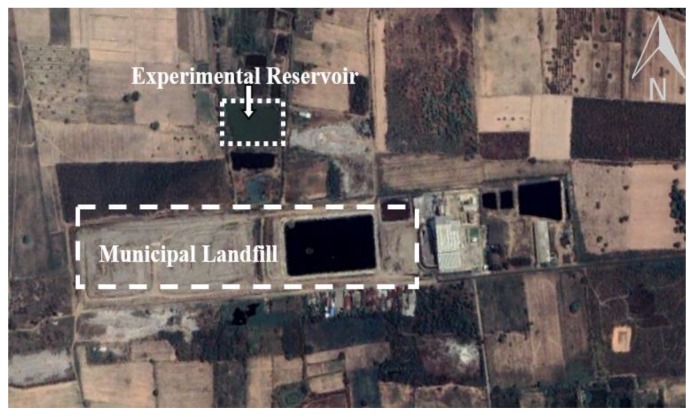
Location of the study site in a metal-contaminated reservoir near the Khon Kaen municipal landfill.

**Figure 2 ijerph-17-01980-f002:**
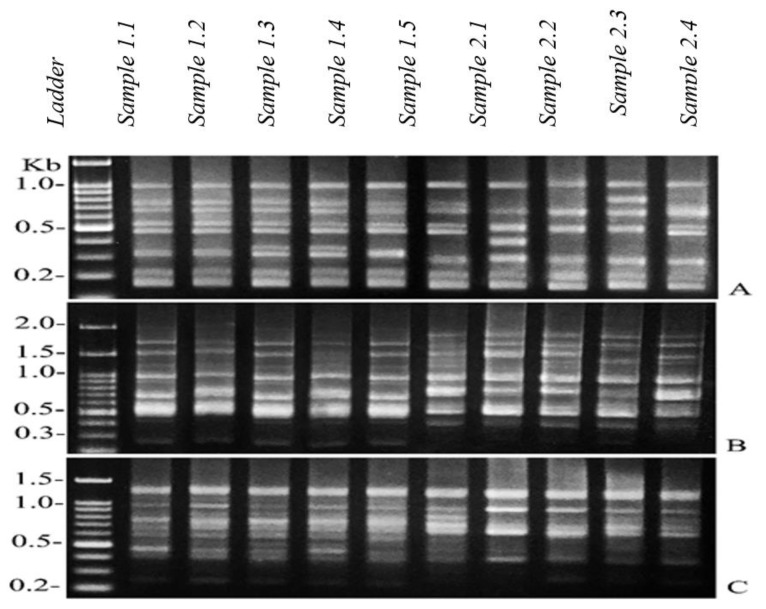
Examples of ISSR fingerprints from the reference catfish (1.1, 1.2, 1.3, 1.4 and 1.5) and the landfill catfish (2.1, 2.2, 2.3, 2.4 and 2.5) from the specific primers TGACCCCTCC. (**A**) GTAGACGAGC, (**B**) TGTCTGGGTG and (**C**) showing monomorphic bands.

**Figure 3 ijerph-17-01980-f003:**
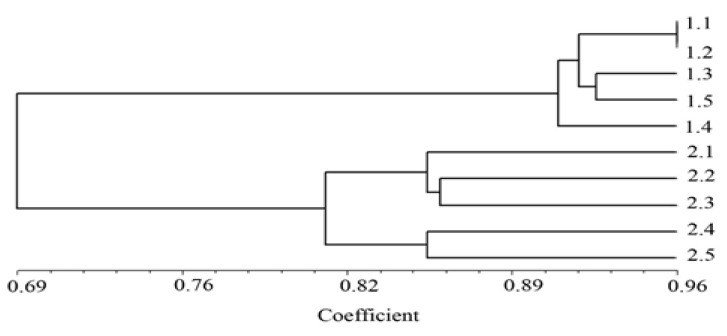
The dendrogram constructed from 25 primers by the NTSYSpc 2.10 program showing genetic relationships of the hybrid catfish samples between the reference catfish (1.1–1.5) and landfill catfish (2.1–2.5).

**Figure 4 ijerph-17-01980-f004:**
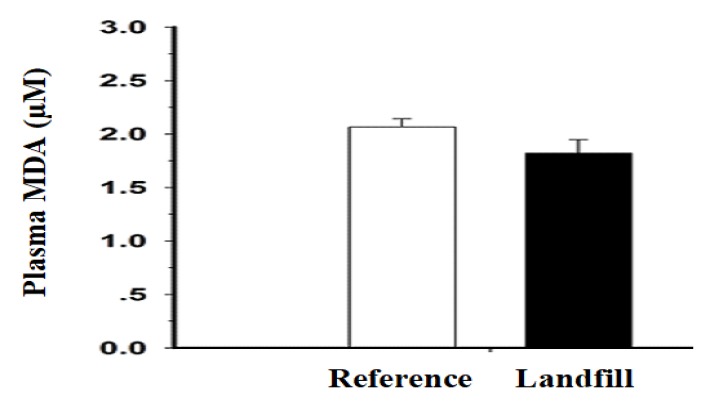
The levels of plasma malondialdehyde (MDA) in both groups of the hybrid catfish. Values are expressed as the mean ± standard deviation of 10 fish.

**Figure 5 ijerph-17-01980-f005:**
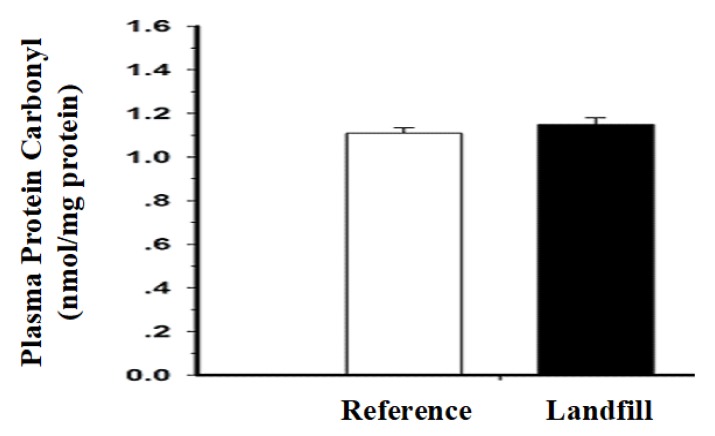
The levels of plasma protein carbonyl (PC) in both groups of the hybrid catfish. Values are expressed as the mean ± standard deviation of 10 fish.

**Table 1 ijerph-17-01980-t001:** Analytical methods used for measurements of water quality parameters.

Water Quality Parameters	Analytical Methods
Dissolved oxygen	DO meter, Model 966, Mettler Toledo
pH	pH meter, Model EcoScan pH 5, Eutech
Temperature	Thermometer
Total dissolved solids	Total dissolved solids, Model CH-8603, Mettler Toledo
Electro-conductivity Total ammonia nitrogen	EC meter, Model CH-8603, Mettler Toledo Titration method

**Table 2 ijerph-17-01980-t002:** The 25 successful primer sequences for inter-simple sequence repeats (ISSRs)-PCR markers.

No.	Nucleotide Sequences	Total Bands	Monomorphic Band	Polymorphic Band
1	CTCTCTCTCTCTCTCTAC	55	1	9
2	CTCTCTCTCTCTCTCTGC	112	7	9
3	CACACACACACAAC	50	1	9
4	CACACACACACAGT	105	5	8
5	CACACACACACAAG	109	7	9
6	CACACACACACAGG	110	5	10
7	GAGAGAGAGAGAGG	79	6	5
8	GAGAGAGAGAGACC	115	7	8
9	CACCACCACGC	110	6	11
10	GAGGAGGAGGC	133	11	5
11	CTCCTCCTCGC	73	6	3
12	GTGGTGGTGGC	103	8	4
13	ACTGACTGACTGACTG	89	3	12
14	GACAGACAGACAGACA	97	8	2
15	CCCCGTGTGTGTGTGT	136	10	7
16	GAGAGAGAGA	105	8	6
17	AGAGAGAGAGAGAGCTGCT	130	10	8
18	CTCTCTCTCTCTCTCTTG	113	7	8
19	AGAGAGAGAGAGAGAA	85	6	5
20	AGAGAGAGAGAGAGAGC	134	8	8
21	AGAGAGAGAGAGAGAGT	152	9	10
22	AGAGAGAGAGAGAGAAA	91	6	7
23	AGAGAGAGAGAGAGAAC	102	5	9
24	AGAGAGAGAGAGAGAAG	110	7	8
25	AGAGAGAGAGAGAGAAT	115	7	8
	Total	2613	164	188

**Table 3 ijerph-17-01980-t003:** Water quality parameters of the reference and the landfill reservoirs.

Samples	Parameters					
	DO (mg/L)	pH	Temperature (°C)	TDS (mg/L)	EC (µSm^−1^/s)	NH_3_-N (mg/L)
Reference reservoir	5.31 ± 0.76	7.02 ± 0.02	25.96 ± 0.55	0.44 ± 0.03	304.33 ± 7.37	0.72 ± 0.04
Landfill reservoir	4.79 ± 0.44	6.72 ± 0.07	26.18 ± 0.53	0.57 ± 0.03	452.67 ± 17.51	0.94 ± 0.04

Remarks: DO = dissolved oxygen; TDS = total dissolved solids; EC = electro-conductivity; NH_3_-N = total ammonia nitrogen; (mean and standard deviation; n = 3).

**Table 4 ijerph-17-01980-t004:** Heavy metal concentrations in the water, sediment and fish muscles from the reference reservoir (mean ± standard deviation; n = 9).

Samples	Individual Number	Cr	Cd	Pb
Water (mg/L)	9	ND	ND	ND
Standard (mg/L)		0.005 ^a^	0.05 ^a^	0.05 ^a^
Sediment (mg/kg)	9	4.95 ± 0.27	0.07 ± 0.01	ND
Standard (mg/kg)		100 ^b^	1 ^b^	100 ^b^
Fish Muscles (mg/kg)	9	0.72 ± 0.12	ND	ND
Standard (mg/kg)		2 ^c^	0.5 ^c^	0.5 ^c^

^a^ Water quality standards for surface water sources, Pollution Control Department, Ministry of Natural Resources and Environment, Thailand [31] ^b^ Soil quality standard, Pollution Control Department, Ministry of Natural Resources and Environment, Thailand [32] ^c^ Thailand’s food quality standards, Ministry of Public Health, Thailand [33].

**Table 5 ijerph-17-01980-t005:** Heavy metal concentrations in the water, sediment and fish muscles from the landfill reservoirs (mean ± standard deviation; n = 9).

Samples	Individual Number	Cr	Cd	Pb
Water (mg/L)	9	16.73 ± 1.32	0.66 ± 0.32	17.85 ± 4.28
Standard (mg/L)		0.005 ^a^	0.05 ^a^	0.05 ^a^
Sediment (mg/kg)	9	33.82 ± 7.79	2.60 ± 1.07	16.61 ± 9.47
Standard (mg/kg)		100 ^b^	1 ^b^	100 ^b^
Fish Muscles (mg/kg)	9	1.41 ± 0.29	0.03 ± 0.01	0.12 ± 0.02
Standard (mg/kg)		2 ^c^	0.5 ^c^	0.5 ^c^

^a^ Water quality standards for surface water sources, Pollution Control Department, Ministry of Natural Resources and Environment, Thailand [31] ^b^ Soil quality standard, Pollution Control Department, Ministry of Natural Resources and Environment, Thailand [32] ^c^ Thailand’s food quality standards, Ministry of Public Health, Thailand [33].

**Table 6 ijerph-17-01980-t006:** The genetic similarity values among the hybrid catfish from the reference (1.1–1.5) and the landfill areas (2.1–2.5), according to the Dice coefficient.

	1.1	1.2	1.3	1.4	1.5	2.1	2.2	2.3	2.4	2.5
1.1	1.00									
1.2	0.96	1.00								
1.3	0.93	0.92	1.00							
1.4	0.90	0.92	0.92	1.00						
1.5	0.92	0.91	0.93	0.91	1.00					
2.1	0.72	0.72	0.70	0.71	0.71	1.00				
2.2	0.69	0.67	0.68	0.68	0.70	0.86	1.00			
2.3	0.71	0.69	0.69	0.68	0.68	0.86	0.86	1.00		
2.4	0.68	0.67	0.66	0.65	0.68	0.80	0.80	0.84	1.00	
2.5	0.69	0.68	0.67	0.66	0.66	0.80	0.79	0.86	0.86	1.00

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
