# Peer review of "Genotoxicity and Oxidative Stress in Experimental Hybrid Catfish Exposed to Heavy Metals in a Municipal Landfill Reservoir"

_ijerph, 2020, doi:10.3390/ijerph17061980_

Round 1

Reviewer 1 Report

It is interesting to explore the effects of practical chronic exposures of heavy metals (Cr, Cd and Pb) on the genotoxicity and oxidative stress in catfish, which is a commonly consumed fish species. However, there are some minor questions and suggestions on this research.

  1. Table 3 & 4 The data is expressed at “mean ± SD” or? Please describe in notes.
  2. Figure 4 and 5 The title of Y axis is better to vertically display and close to the Y axis for Figure 4 and 5. And the legends in figure is not necessary because they already show in the X axis title.
  3. Figure 4 and 5 Because the data is from 10 catfishes instead of repeated experiments, “mean ± SD” should be used instead of “mean ± SEM”. But because Mann-Whitney U-test is used for the comparisons of the plasma MDA and PC levels, if the data is abnormal distribution, it is better to show the median and quartiles.
  4. Line 94 When was the collect time for water sample or the detect time for water quality parameters?  At the end of the experiment? Please also describe in method section.
  5. Lin 101-112 As the exposure time of catfish is 4 months, when and how to collect the water and sediment samples? Please describe the sampling method, frequency etc.
  6. Table 4 It is only shown the heavy metal concentrations of fish muscle from the landfill reservoir. How about the concentrations of fish muscle from reference reservoir? If possible, it is better to detect or report, too, even if it might be lower than the detection limits.
  7. Line 150 As Line 90 and 114 described, it was analysed the DNA of the gill and liver of catfish. What organ was the results of genotoxicity? From the catfish gill, liver, or their mixture?

Author Response

Please kindly consider the attached response following questions and suggestions related to the article 725537, and becomes acceptable for publishing in ijerph.

Best regards,

Bundit Tengjaroenkul

Reviewer 2 Report

The paper is very well structured, but there are two mayor features to be considered:

  1. According to IUPAC, the term “Heavy metals” is imprecise and misleading; no standard definition of this term currently exists. Researchers should only use well-accepted definitions.Duffus, J.H. Pure Appl. Chem., Vol. 74, No. 5, pp. 793–807, 2002Pourret O. and Hursthouse A. Int. J. Environ. Res. Public Health 2019, 16, 444
  2.  
  3. Pourret O. Sustainability 2018, 10, 2879
  4. I suggest the following readings:
  5. In line 246 you affirm that “the genetic differentiation in the hybrid catfish revealed a positive correlation with Cr, Cd and Pb concentrations accumulated in the fish body”, but you don´t show any data that explain such correlation.

You show the results of the analysis of Cr, Cd, and Pb in fish muscle from the landfill reservoir and confirm that they don´t exceed the Thailand food quality standard, in spite of the concentrations of these elements in the water. I suggest that you should analyse the concentration of Cr, Cd and Pb in fish muscle from the reference reservoir, if you find differences between Cr, Cd, Pb in fish muscle from landfill reservoir versus reference reservoir, maybe you could get valuable conclusions.

Minor:

Line 108: I don´t understand quite well, I wonder if “and standards” should not be here.

Line 211: The concentration of Cd in sediment is 2.60 µg/g (above 1 µg/g), but the uncertainty of the result (sd 2.07 µg/g) needs to be considered.

Lines 263-279: On one hand, at the beginning of the paragraph, no significant differences are observed between the MDA analysis of two populations; on the other hand, at the end of the paragraph (line 276-279), the decrease in plasma MDA level of one of the population could be the reason of strengthen cellular structures or metabolic processes….

In my opinion, if in the results of the analysis of a biological/chemical property of two groups there have been found that there aren´t significant differences, the conclusion is that, there are no differences between both groups according to such property.

Line 286: the expression “slightly greater level” should be supported by data, and should be explained if they are significantly different according to the analytical technique used.

Author Response

Please kindly consider the response related to questions and suggestions of the article 725537, and become acceptable for publishing in ijerph.

Best regards,

Bundit Tengjaroenkul

Round 2

Reviewer 2 Report

Some consideration have been corrected, but there are relevant ideas to be considered:

The suggestion of reading the papers was to understand why it should be avoided the use of the term “Heavy Metal”, not to just include them as references and keep using the expression “Heavy Metal”.

The authors analyse MDA and PC as Oxidative Stress Biomarkers in two populations, landfill reservoir and reference reservoir. They found that there are not statistically significant differences between the two groups, which means that there are no differences between both groups according to MDA and PC, so the conclusion is that there is not Oxidative Stress in landfill reservoir, according to this experiment.

The authors have edited in the section 4.3 Genotoxicity, lines 250-251: “The genetic differentiation in the hybrid catfish revealed a decrease its value with the increase Cr, Cd and Pb concentrations accumulated in the fish body”. The sentence is not quite well understandable. Maybe it is not grammatically correct: “a decrease its values”. Which values?

Line 33: There are no data of arsenic in the paper

Author Response

Dear Reviewer,

Comments and suggestions were edited as in the yellow highlight text.

Please consider this article as accepted for publication in Journal Environmental Research and Public Health.

Best regards,

Bundit Tengjaroenkul

Round 3

Reviewer 2 Report

Dear Authors and Editor,

In Reviews 1 and 2, I have suggested not using the term “Heavy Metals” because according to IUPAC, this term is imprecise and misleading; no standard definition of this term currently exists and researchers should only use well-accepted definitions. (Duffus, J.H. Pure Appl. Chem., Vol. 74, No. 5, pp. 793–807, 2002).

Even I have suggested the reading of two recent papers, one of them published in Int. J. Environ. Res. Public Health, that support IUPAC recommendation (Pourret O. Sustainability 2018, 10, 2879; Pourret O. and Hursthouse A. Int. J. Environ. Res. Public Health 2019, 16, 444). However, the authors keep using the term in the paper.

At this point, I leave to the judgment of the Editor if the paper is accepted or not for publishing in the journal Int. J. Environ. Res. Public Health.

Author Response

Dear reviewer 2,

Thank you for all of your comments and suggestions.

Now, everything is up to the editor as in your judgment.

I hope, it will turn out to be positive, if the article is suitable enough.

Best regards,

Bundit Tengjaroenkul
